# Pathogenic Mechanisms of Metabolic Dysfunction-Associated Steatotic Liver Disease (MASLD)-Associated Hepatocellular Carcinoma

**DOI:** 10.3390/cells14060428

**Published:** 2025-03-13

**Authors:** Toru Nakamura, Atsutaka Masuda, Dan Nakano, Keisuke Amano, Tomoya Sano, Masahito Nakano, Takumi Kawaguchi

**Affiliations:** 1Division of Gastroenterology, Department of Medicine, School of Medicine, Kurume University, Kurume 830-0011, Japan; ntoru@kurume-u.ac.jp (T.N.); masuda_atsutaka@med.kurume-u.ac.jp (A.M.); nakano_dan@med.kurume-u.ac.jp (D.N.); amano_keisuke@kurume-u.ac.jp (K.A.); sano_tomoya@med.kurume-u.ac.jp (T.S.); nakano_masahito@kurume-u.ac.jp (M.N.); 2Liver Cancer Research Division, Research Center for Innovative Cancer Therapy, Kurume University, 67 Asahi-Machi, Kurume 830-0011, Japan; 3Fukuoka Consulting and Support Center for Liver Diseases, Kurume 830-0011, Japan

**Keywords:** hepatocellular carcinoma, hepatoma, liver cancer, metabolic dysfunction-associated steatotic liver disease, fatty liver, steatosis, inflammation, SGLT2 inhibitor, CD34-positive cells, myokines

## Abstract

Hepatocellular carcinoma (HCC) is the sixth most common cancer and the third leading cause of cancer deaths worldwide. The etiology of HCC has now dramatically changed from viral hepatitis to metabolic dysfunction-associated steatotic liver disease (MASLD). The main pathogenesis of MASLD-related HCC is the hepatic lipid accumulation of hepatocytes, which causes chronic inflammation and the subsequent progression of hepatic fibrosis. Chronic hepatic inflammation generates oxidative stress and DNA damage in hepatocytes, which contribute to genomic instability, resulting in the development of HCC. Several metabolic and molecular pathways are also linked to chronic inflammation and HCC in MASLD. In particular, the MAPK and PI3K-Akt-mTOR pathways are upregulated in MASLD, promoting the survival and proliferation of HCC cells. In addition, MASLD has been reported to enhance the development of HCC in patients with chronic viral hepatitis infection. Although there is no approved medication for MASLD besides resmetirom in the USA, there are some preventive strategies for the onset and progression of HCC. Sodium-glucose cotransporter-2 (SGLT2) inhibitor, a class of medications, has been reported to exert anti-tumor effects on HCC by regulating metabolic reprogramming. Moreover, CD34-positive cell transplantation improves hepatic fibrosis by promoting intrahepatic angiogenesis and supplying various growth factors. Furthermore, exercise improves MASLD through an increase in energy consumption as well as changes in chemokines and myokines. In this review, we summarize the recent progress made in the pathogenic mechanisms of MASLD-associated HCC. Furthermore, we introduced new therapeutic strategies for preventing the development of HCC based on the pathogenesis of MASLD.

## 1. Etiology of Hepatocellular Carcinoma (HCC)

### 1.1. From Viral HCC to Non-Viral HCC

HCC is the sixth most prevalent cancer and the third leading cause of cancer-related deaths globally. In Japan, viral hepatitis continues to be the primary cause of HCC, although the proportion of cases linked to hepatitis C virus (HCV) has declined [1,2]. However, Tateishi et al. found that, between 1991 and 2010, the incidence of non-viral HCC increased significantly, with a higher proportion of patients having obesity or diabetes compared to those with viral hepatitis [3]. We also analyzed the epidemiological trends of HCC incidence in Japan over the past 24 years (1996–2019) [4]. In this study, data from 20,547 HCC patients were analyzed. We included only those newly diagnosed HCC patients in this study who were diagnosed at 1 of the 19 participating institutions of the Liver Cancer Study Group of Kyushu between 1996 and 2019. The distribution of factors such as sex, age, and disease etiology were examined among the new HCC cases. Metastatic liver cancer was an exclusion criterion in this study. The etiology of HCC was classified into four categories: hepatitis B virus (HBV) infection, HBV + HCV co-infection, hepatitis C virus (HCV) infection, and non-viral (HBV and HCV negative). HCC cases were attributed to HBV, HBV + HCV, HCV, and non-viral causes in 2997 (14.6%), 187 (0.9%), 12,019 (58.5%), and 5344 (26.0%) patients, respectively. While the proportion of hepatitis virus-associated HCC has declined, the incidence of non-viral HCC has increased (Figure 1). Recent studies by Tateishi et al., Shiomi et al., and Sasaki et al. have highlighted the role of diabetes mellitus in the development of non-viral HCC [5,6,7]. Furthermore, steatotic liver disease (SLD) is the leading cause of chronic liver disease, affecting approximately 25% of the global adult population [8,9,10]. SLD can result in significant hepatic fibrosis, elevating the risk of HCC and mortality [11,12,13]. As a result, the incidence of HCV-related HCC has significantly declined. Conversely, the incidence of non-viral HCC has been rapidly increasing, particularly among patients with metabolic dysfunctions.

Given the decreasing incidence of HCV-related HCC and the rapid rise in non-viral HCC associated with metabolic dysfunctions, future efforts should focus on understanding the impact of metabolic factors, such as diabetes and steatotic liver disease, on HCC development and on developing effective prevention and management strategies.

### 1.2. From NAFLD to MASLD

In 2023, the Non-Alcoholic Fatty Liver Disease (NAFLD) Nomenclature Consensus Group released a multi-society Delphi consensus statement introducing a new terminology for fatty liver disease. The term selected to replace NAFLD was metabolic dysfunction-associated steatotic liver disease (MASLD) [14,15,16]. MASLD is characterized by hepatic steatosis accompanied by at least one of five cardiometabolic risk factors [14]. MASLD is the leading cause of chronic liver disease (CLD) and can progress to hepatic fibrosis, increasing the risk of HCC and mortality in Asia [11,12,13].

Evaluating hepatic fibrosis and steatosis is crucial for the diagnosis and staging of MASLD. Non-invasive tests (NITs) are the most commonly used method for assessing NAFLD [17,18] and are alternatives to liver biopsy for identifying patients with NAFLD who are at high risk [19,20]. In particular, advanced hepatic fibrosis (≥F2) is linked to hepatocarcinogenesis and affects prognosis [21]. However, it remains uncertain whether the NITs for hepatic fibrosis and steatosis used in NAFLD are equally effective for MASLD. To address this, we compared hepatic fibrosis and steatosis using NITs between NAFLD and MASLD in Asia [22].

We examined 1907 consecutive health checkup participants from May 2017 to September 2022. Shear-wave elastography (SWE) was conducted during routine ultrasound exams, and patients with liver stiffness ≥ 6.60 kPa were considered to have marked hepatic fibrosis. Among these individuals, 647 patients met the criteria for NAFLD diagnosis, and 640 for MASLD, resulting in a 99% overlap. Regarding the NITs for hepatic fibrosis and steatosis, the values and prevalence of advanced fibrosis were nearly identical. Therefore, 99% of patients with NAFLD also met the MASLD criteria in Asia. Ratziu et al. noted that at least 98% of patients in the current NAFLD database would be classified as having MASLD in Europe [23,24]. Our findings indicate that NITs for hepatic fibrosis and steatosis used in NAFLD are also effective for MASLD in Asia. Additionally, several recent studies have shown that MASLD shares similar clinical characteristics with NAFLD, including hepatic fibrosis, the prognosis of HCC and gastrointestinal cancers, incidence of atherosclerotic cardiovascular disease, reflux esophagitis, chronic obstructive pulmonary disease, and patient-reported outcomes [22,25,26,27,28,29,30,31,32,33].

Recently, Qiu et al. conducted a cross-sectional study to examine the increasing trend in HCC [34]. They analyzed data from the National Vital Statistics System, which includes 188,280 HCC-related deaths between 2006 and 2022. The study found that age-standardized mortality rates decreased for HCV- and HBV-related deaths, but increased for MASLD-related deaths. Moreover, the number of MASLD-related HCC deaths rose rapidly from 2006 to 2022, with MASLD surpassing HBV as the third-leading cause of HCC-related death. It is projected that MASLD will overtake HCV by 2032, becoming the second-leading cause of HCC-related death [34].

We will focus on evaluating the utility of non-invasive tests (NITs) for hepatic fibrosis and steatosis in MASLD, given their proven effectiveness in NAFLD, and further investigate the clinical implications of MASLD, including its association with hepatic fibrosis, hepatocarcinogenesis, and overall prognosis in Asia.

### 1.3. HCC Subclass

Significant effort has been dedicated to classifying HCC at the molecular, metabolic, and immunological levels [35]. The pathogenesis of HCC is highly complex and involves multiple molecular disruptions, including cell cycle deregulation, alterations in DNA methylation, chromosomal instability, immunomodulation, epithelial-to-mesenchymal transition, an increase in HCC stem cells, and dysregulation of microRNA [36,37]. While the specific disease mechanisms vary depending on the underlying etiology, the typical progression follows a sequence of liver injury, chronic inflammation, fibrosis, cirrhosis, and, ultimately, HCC [35].

HCC consists of a diverse mix of cell types embedded within the extracellular matrix and supportive stroma, including malignant hepatocytes, immune cells, and endothelial cells. Sia et al. reported that an analysis of HCC samples from 956 patients revealed that nearly 25% exhibited markers of an inflammatory response [38]. They identified two distinct subclasses: one characterized by an adaptive immune response and the other by an exhausted immune state [38]. These findings suggest that certain HCCs may be responsive to therapeutic agents targeting regulatory pathways in T cells, such as inhibitors of programmed death-ligand 1, programmed cell death 1, or transforming growth factor beta 1 inhibitors.

Growing research evidence indicates that HCC is a highly heterogeneous disease, with genetic profiling revealing multiple distinct subtypes. The canonical Wnt/β-catenin pathway, a complex and evolutionarily conserved signaling mechanism, plays a crucial role in HCC development and progression [39]. Notably, the WNT-high subtype is linked to an immunosuppressive microenvironment, poor prognosis, activation of cancer-related pathways, and limited responsiveness to immune checkpoint therapy. The Wnt signaling-based classification for HCC is expected to have significant clinical implications for both prognosis assessment and immunotherapy selection.

MASLD-related HCC is closely associated with chronic inflammation, which plays a critical role in its development and progression. Among the molecular subtypes of HCC, the inflammation-related subclass identified by Sia et al. is particularly relevant [38]. Given that MASLD is associated with persistent metabolic inflammation, MASLD-derived HCC is likely linked to the immune-low subtype, which features sustained immune suppression and chronic inflammatory signaling. Additionally, the Wnt/β-catenin signaling pathway is believed to contribute to MASLD-related HCC [39]. Since MASLD is associated with metabolic dysfunction, fibrosis, and steatosis, dysregulation of the Wnt/β-catenin pathway may play a significant role in forming the immunological landscape of MASLD-related HCC. These classifications highlight the complex interactions between chronic metabolic inflammation, immune suppression, and tumor progression in MASLD-derived HCC.

## 2. Mechanisms of MASLD-Related HCC: Inflammation as a Key Factor

Chronic liver inflammation is central to the progression from MASLD to HCC, with inflammatory pathways causing sustained liver injury, fibrosis, and carcinogenesis. The following outlines key mechanisms through which inflammation contributes to HCC in MASLD (Table 1).

### 2.1. Lipid-Accumulation-Induced Chronic Inflammation

The initial stages of MASLD often involve an accumulation of lipids in hepatocytes, arising from metabolic imbalances such as insulin resistance and obesity-related stress [40,41]. This lipid excess triggers hepatocyte stress responses, resulting in the recruitment of liver-resident immune cells, including Kupffer cells and hepatic stellate cells (HSCs). Kupffer cells detect signals from stressed hepatocytes and respond by releasing pro-inflammatory cytokines such as TNF-α and IL-6. These cytokines create a pro-inflammatory environment, attracting more immune cells and perpetuating a cycle of chronic low-grade inflammation [42]. Over time, this sustained inflammatory response damages hepatocytes further, setting the stage for fibrotic changes and potential carcinogenesis. In this way, inflammation serves as a direct consequence of lipid accumulation, transforming metabolic stress into persistent immune activation.

### 2.2. Oxidative Stress and DNA Damage Accumulation

Inflammation in MASLD also generates substantial oxidative stress in the liver, primarily due to the overproduction of reactive oxygen species (ROS) and reactive nitrogen species (RNS) [43]. The release of ROS and RNS during inflammatory responses causes lipid peroxidation, mitochondrial dysfunction, and DNA mutations in hepatocytes, all of which collectively contribute to genomic instability [44]. ROS and RNS are produced by different immune cell populations in response to liver inflammation [45]. Notably, RNS are predominantly generated by activated neutrophils, which play a crucial role in the onset and progression of MASLD [46]. The excessive production of RNS amplifies oxidative stress, further exacerbating cellular damage and perpetuating the cycle of inflammation and injury. This process not only accelerates the progression of MASLD, but also creates a microenvironment favorable for hepatocarcinogenesis. By inducing oxidative modifications in DNA, proteins, and lipids, RNS contributes to the chronic liver damage that underpins the transition from MASLD to HCC [44,47]. As mutations accumulate in hepatocytes, the risk of these cells acquiring cancer characteristics rises, allowing for them to bypass the liver’s natural cancer-protective mechanisms and progress toward HCC. Thus, understanding the specific roles of ROS and RNS in liver inflammation offers potential therapeutic targets for interrupting the progression of HCC.

### 2.3. Role of the Immune System in Sustaining Inflammation

The immune system—encompassing both innate and adaptive immunity—is central to sustaining inflammation and driving HCC in MASLD. Chronic liver inflammation attracts various immune cells, particularly macrophages and T cells, into hepatic tissue, compounding liver injury [48,49]. Macrophages, once activated, release an array of pro-inflammatory cytokines and chemokines, escalating the immune response. Additionally, the adaptive immune system is involved in the development of HCC through the activation of T lymphocytes. Although cytotoxic T cells are initially beneficial in controlling hepatocyte injury, prolonged immune activation leads to unintended consequences, with these cells inadvertently worsening liver inflammation. This prolonged immune response creates a feedback loop that damages hepatocytes and fosters an environment conducive to HCC development. The presence of sustained immune activity increases cellular injury, driving further damage and increasing the risk of HCC [50].

### 2.4. Fibrosis as a Precursor to Hepatocarcinogenesis

One of the most critical factors predisposing MASLD patients to HCC is fibrosis, primarily driven by chronic inflammation. In MASLD, the persistent inflammatory state activates HSCs, which are instrumental in fibrogenesis due to their role in extracellular matrix (ECM) production [51]. Once activated, HSCs transform into myofibroblast-like cells, resulting in the excessive deposition of ECM proteins. This process gradually leads to fibrosis, and, if left untreated, progresses to cirrhotic liver—a key risk factor for HCC. Liver cirrhosis marks a pivotal stage where hepatocytes are exposed to chronic oxidative stress and DNA damage, conditions that promote genomic instability. This unstable cellular environment elevates the likelihood of oncogenic mutations, thereby increasing the risk of HCC development [42,52]. Thus, fibrosis acts as a transitional phase in which persistent inflammation and ECM deposition ultimately predispose the liver to malignant transformation.

### 2.5. Metabolic Pathways in Inflammation and Carcinogenesis

Several metabolic and molecular pathways have been identified as critical links between inflammation and HCC in MASLD. Notably, the MAPK and PI3K-Akt-mTOR pathways are frequently upregulated in MASLD, promoting the survival and proliferation of abnormal cells [53,54]. Chronic inflammation and oxidative stress activate these pathways in tandem, resulting in uncontrolled cell growth and resistance to apoptosis. Furthermore, upregulation of these pathways not only promotes cell survival, but also enhances the inflammatory cycle and maintains an optimized environment for HCC development. This synergy between inflammation and oncogenic signaling highlights the importance of metabolic dysregulation in driving HCC in MASLD [55].

Recently, Xu et al. found that TRAF6-binding protein (T6BP) is a novel and critical suppressor of protein tyrosine kinase 2 beta (PYK2) that reduces hepatic lipid accumulation, pro-inflammatory factor release, and pro-fibrosis production [56]. T6BP directly targets PYK2 and prevents its dimerization, disrupting downstream PYK2-JNK signaling hyperactivation. Additionally, T6BP favorably recruits ubiquitin ligase targeting PYK2, to form a complex and degrade PYK2. This inhibits the progression of MASH, metabolic dysfunction-associated steatotic liver disease (MASLD)-related HCC (MASH-HCC). In addition, Gilglioni et al. reported that increased expression of hepatic protein tyrosine phosphatase receptor type K (PTPRK) in patients with MASLD [57]. Elevated hepatic PTPRK expression triggers increased glycolysis, culminating in the activation of peroxisome proliferator-activated receptor γ and the stimulation of de novo lipogenesis. They also found that silencing PTPRK in liver cancer cell lines reduces colony-forming capacity and high-fat-fed PTPRK knockout mice exposed to a hepatic carcinogen develop smaller tumors. Thus, T6BP and PTPRK are new pathogenesis of MASLD-related HCC and are considered to be therapeutic targets.

The progression from MASLD to HCC is a multifaceted process, heavily influenced by chronic inflammation. This inflammation drives disease progression by triggering immune pathways, activating fibrogenic processes, inducing oxidative stress, and sustaining oncogenic molecular pathways. Through these mechanisms, inflammation transforms the liver microenvironment, creating a breeding ground for cancer development. Targeting these inflammatory pathways may offer new therapeutic avenues for reducing HCC risk in MASLD patients.

### 2.6. Liver Microenvironment and Pro-Carcinogenic Signaling Pathways

Persistent inflammation in MASLD alters the liver microenvironment, enriching it with cytokines and growth factors such as TGF-β and IL-6, which collectively create conditions that support tumor growth [42]. These factors activate key signaling pathways, including the JAK/STAT and NF-κB pathways, both of which encourage hepatocyte proliferation and survival [58]. Continuous activation of these pathways perpetuates inflammation, prevents the natural process of apoptosis, and supports the survival of cells with mutations. This cascade of events accelerates the progression from fibrosis to carcinogenesis, as pro-carcinogenic signals within the microenvironment enable mutated hepatocytes to evade normal growth regulation. Over time, these cells continue to proliferate, adapting and surviving in the adverse inflammatory environment, advancing toward malignancy.

In summary, it is important to further elucidate the role of inflammation in the pathogenesis of MASLD-related HCC and explore potential therapeutic targets. Basic research needs to investigate in detail the functional mechanisms of novel factors such as T6BP and PTPRK and the regulation of the JAK/STAT and NF-κB pathways. Additionally, clinical research should address the prediction of HCC risk using blood biomarkers and the evaluation of the efficacy of anti-inflammatory therapies. The development of novel inflammation-targeted therapies will broaden the possibilities for prevention and treatment of MASLD-related HCC.

## 3. Impact of MASLD on HCC in Patients with HBV and HCV

There has been notable global progress towards eliminating viral hepatitis [59,60,61]. The treatment of HBV has seen remarkable advancements with nucleos(t)ide analogs (NAs) [62,63]. These medications suppress HBV replication and decrease the risk of liver disease progression [64,65,66]. Similarly, the use of direct-acting antivirals (DAAs) against HCV has enabled sustained virological response (SVR) in almost all cases [67,68,69].

Lee et al. reported the impact of MASLD on the development of cirrhosis and HCC by chronic hepatitis B or C infection and antiviral drug treatment status in Taiwan. Among patients with chronic hepatitis B virus infection or HCV infection who received antivirals during follow-up, MASLD was associated with increased risks of cirrhosis and HCC, with adjusted HRs of 1.23 (95% CI, 1.01–1.49) and 1.32 (95% CI, 1.05–1.65), respectively [70].

In patients with chronic HBV infection, Adali et al. showed that the cumulative incidence of HCC increased with the number of fulfilled cardiometabolic criteria (0–2 criteria vs. ≥3 criteria) (HR 3.93; 95% CI 1.89–8.19; *p* < 0.001) [71]. Similarly, we reported that SLD with metabolic dysfunction was an independent factor associated with the onset of HCC in HBeAg-negative patients with undetectable HBV-DNA using NAs therapy (OR 2.4, 95% CI 1.0–6.0, *p* = 0.04) [72]. In contrast, Huang et al. reported that HBV-infected patients with concurrent steatosis tended to have lower viral activity, including lower serum HBV DNA levels and higher rates of hepatitis B surface antigen seroclearance [73]. Therefore, the influence of co-existing steatosis liver disease in HBV-infected patients remained controversial. Although MASLD and chronic hepatitis B are well-established etiologies for HCC, whether concurrent MASLD and chronic HBV infection lead to a higher risk of HCC development than HBV alone is inconclusive.

In patients with chronic HCV infection, Liu et al. reported that patients with MASLD exhibited an increased HCC risk (adjusted hazard ratio 2.07; 95% CI 1.36–3.16) compared to those without MASLD after achieving SVR. Vigilant HCC surveillance and control of cardiometabolic risk factors to mitigate the effect of MASLD on HCC remain crucial for this population [74]. We also reported the incidence and risk factors of HCC after DAA treatment in a large multicenter cohort in Japan [75]. Of the 2552 patients achieving SVR through DAA treatment, 70 (2.7%) developed HCC. The 12-, 24-, and 36-month cumulative HCC incidences were 1.3%, 2.9%, and 4.9% in all patients; 2.5%, 5.2%, and 10.0% in those with cirrhosis; and 0.9%, 2.1%, and 2.9% in those without cirrhosis, respectively. In addition to the previous report, we showed the effect of MASLD on the development of HCC after SVR [76]. We enrolled 1280 elderly patients with HCV eradication with DAA treatment. The exclusion criteria were as follows: (i) history of HCC before DAA treatment, (ii) HCC development before SVR. In total, 86 patients (6.7%) developed HCC during the follow-up period (35.8 ± 23.7 months). In the multivariate analysis, serum α-fetoprotein level (HR 1.08, CI 1.04–1.11, *p* = 0.0008), FIB-4 index (HR 1.17, CI 1.08–1.26, *p* = 0.0007), and MASLD (HR 3.04, CI 1.40–6.58, *p* = 0.0125) at 24 weeks of SVR were independent factors associated with HCC development. Therefore, MASLD increases the risk of HCC, even in patients with HCV infection after the achievement of SVR. In the future, even when viral hepatitis is already under control, it is important to be aware of the complications of MASLD and to continue to manage it with images and biomarkers (Figure 2).

## 4. Preventive Strategies for the Onset and Progression of HCC

Recently, resmetirom, a thyroid hormone receptor-β agonist, was approved in the USA [77]. In addition, the following compounds are under clinical trials and are good candidates for MASLD: GLP-1 receptor agonist, a dual agonist of glucagon receptor and GLP-1 receptor; pan-PPAR agonist; and FGF21 analog [78,79,80,81,82,83,84]. However, resmetirom is not available elsewhere, and these new candidates have not been approved yet. Therefore, it is necessary to take measures to prevent the onset and progression of HCC.

### 4.1. Sodium-Glucose Cotransporter-2 Inhibitors (SGLT2i)

Type 2 diabetes mellitus is frequently seen in patients with MASLD and is a potent risk factor for HCC [85,86]. Therefore, treatment for type 2 diabetes mellitus has the potential to prevent the onset of HCC, especially in MASLD-related HCC [87]. The clinical practice guidelines for NAFLD/Metabolic dysfunction-associated fatty liver disease (MAFLD)/MASLD of several major societies of hepatology recommended the following anti-diabetic medications for NAFLD/MAFLD/MASLD patients with type 2 diabetes mellitus: thiazolidinediones, glucagon-like peptide-1 receptor agonist, and SGLT2I [88,89,90,91].

SGLT2 inhibitors prevent the reabsorption of glucose from the tubular lumen by acting on the SGLT2 proteins expressed in the proximal tubules in the kidney. In addition to its glucose-lowering properties, SGLT2 inhibitors have shown cardiovascular and renal benefits in numerous large-scale randomized clinical trials, even among patients without type 2 diabetes mellitus [92,93,94]. In addition, SGLT2i has been reported to improve NAFLD/MAFLD/MASLD [95,96,97,98,99,100,101,102,103,104,105].

Insulin resistance is a feature of MASLD, and SGLT2i improves insulin resistance [106,107,108,109]. Insulin resistance is involved in the accumulation of triglycerides in hepatocytes, hepatic inflammation, and hepatic fibrosis [110,111,112]. Qiang et al. investigated the effects of SGLT2i, luseogliflozin, on nonalcoholic steatohepatitis (NASH) development using a rodent model [113]. They found that luseogliflozin suppressed hepatic lipid accumulation and decreased serum alanine aminotransferase compared to those in the control group. In addition, luseogliflozin attenuated fibrotic change and increased collagen deposition with upregulations of collagen1, smooth-muscle actin, and inflammatory cytokine expressions. Honda et al. also reported that Ipragliflozin, an SGLT2i, decreased the serum levels of free fatty acids, hepatic lipid content, the number of apoptotic cells, and areas of fibrosis [114]. Furthermore, Jojima et al. reported that empagliflozin, an SGLT2i, suppressed hepatic expression of inflammatory genes (tumor necrosis factor-α, interleukin-6, and monocyte chemoattractant protein-1) compared to the vehicle group. Immunohistochemistry showed that treatment with empagliflozin reduced the expression of α-smooth-muscle actin compared with the vehicle group [115]. In clinical studies, Takahashi et al. performed a multicenter randomized controlled trial to investigate the effects of ipragliflozin on NAFLD using liver biopsy [98]. They found that NASH resolution and amelioration of hepatic fibrosis were observed in between 66.7% and 70.6% of the ipragliflozin group, respectively. In addition, we recently performed a pooled meta-analysis of phase III clinical trials and demonstrated that 24-week treatment with luseogliflozin improved the hepatic steatosis and fibrosis indexes in diabetic patients, especially those with liver injury (Figure 3). This study has several limitations. Firstly, all phase III clinical trials were conducted in Japan. Secondly, the evaluation of hepatic steatosis and fibrosis relied on non-invasive biological tests. Thirdly, the treatment duration was 24 weeks. Therefore, future research should be structured as a meta-analysis of clinical trials conducted in various regions, incorporating data from imaging modalities or liver biopsies, and extending the treatment periods. However, the accumulated basic and clinical evidence demonstrates that SGLT2i improves the accumulation of triglycerides in hepatocytes, hepatic inflammation, and the hepatic fibrosis of MASLD, which are risk factors for HCC.

SGLT2i has recently been reported to exert anticancer effects in patients with various cancers [102,117,118,119,120,121,122,123,124,125,126,127]. Basic studies have shown that the mechanisms of these beneficial effects are based on improvements in metabolic dysfunctions. Obara et al. reported that tofogliflozin suppresses chronic inflammation and hepatic lipidosis in C57BL/KsJ−/+Lepr db/+Lepr db obese and diabetic mice, thereby reducing the early stages of HCC associated with obesity and MASLD [128]. Moreover, Kaji et al. reported that canagliflozin, an SGLT2i, suppresses cell growth in HCC by inhibiting ATP production, cell cycle, and angiogenesis [129]. In addition, Shiba et al. reported that canagliflozin attenuates the development of HCC through healthy adipose expansion in Western diet-fed melanocortin 4 receptor-deficient mice, a mouse model of human NASH [130]. In addition, we investigated the effects of canagliflozin on the proliferation and metabolic reprogramming of HCC cell lines using a multi-omics analysis of metabolomics and absolute quantification proteomics (iMPAQT) [131]. We found that canagliflozin suppressed the proliferation of HCC cells through alterations in mitochondrial oxidative phosphorylation metabolism, fatty acid metabolism, and purine and pyrimidine metabolism (Figure 4). Thus, canagliflozin directly suppresses the proliferation of HCC by regulating metabolic reprogramming.

We experienced a case of regression of HCC with downregulation of angiogenesis-related cytokines after treatment with SGLT2i in a cirrhotic patient with diabetes mellitus [118]. In various clinical studies, the beneficial effects of SGLT2i have also been reported. Chou et al. conducted a retrospective cohort study of patients with type 2 diabetes mellitus in Hong Kong. They demonstrated that SGLT2i use was associated with a lower risk of HCC compared to dipeptidyl peptidase 4 inhibitor (DPP4i) use after adjustments and in the context of cirrhosis, advanced fibrosis, HBV infection, and HCV infection [122]. Mao et al. performed a population-based cohort study and demonstrated that SGLT-2i users were associated with a substantially lower risk of liver and non-liver complications than other glucose-lowering drug users among patients with diabetic MASLD [132]. They also showed that the risk was further reduced with concomitant metformin use. Chung et al. examined the association between SGLT2 inhibition and long-term liver-related complications using Mendelian randomization and long-term follow-up of a population-based cohort using two European cohorts and a Korean nationwide population-based cohort study [133]. They found that SGLT2 inhibition is associated with a lower risk of liver-related events. We also investigated the impact of SGLT2i on the incidence of extrahepatic cancer compared to DPP4i in patients with type 2 diabetes mellitus and suspected MASLD using a medical claims database in Japan (Figure 5) [102]. We enrolled 1,628,656 patients with type 2 diabetes mellitus who were prescribed either SGLT2i or DPP4i between 1 April 2014 and 31 October 2022. The inclusive criteria were as follows: (i) prescription of either SGLT2i or DPP4i; (ii) no prescription of the comparator drug or its combination within 6 months before the index day; (iii) available medical records from 6 months before the prescription and a confirmed diagnosis of type 2 or unspecified diabetes mellitus. Exclusion criteria were as follows: (i) diagnosis of type 1 diabetes mellitus, malnutrition-related diabetes, other specified diabetes, or diabetes in pregnancy during the database period; and (ii) diagnosis of any disease associated with abnormal liver function tests, except for MASLD/MASH, within 6 months before the index date. We found that SGLT2i significantly suppressed the incidence of extrahepatic cancer (HR 0.50, 95%CI 0.30–0.84, *p* = 0.009) compared to DPP4i. Extrahepatic cancer, along with HCC and major adverse cardiovascular events, is one of the leading causes of death in patients with MASLD. Thus, SGLT2i may improve the prognosis of patients with MASLD through the inhibition of HCC and extrahepatic cancer. This study has several limitations. Firstly, it was an observational study, so we cannot rule out the presence of unassessed confounding factors. Secondly, the database lacked information on body mass index, abdominal circumference, and imaging data. Consequently, we defined suspected MASLD using a predefined ALT cutoff for Japanese patients with NAFLD and excluded other liver diseases. Hepatic fibrosis was assessed using non-invasive indices rather than liver biopsy. Thirdly, the observational period was 12 months, and the incidence of liver-related events, cardiovascular events, and extrahepatic cancer needs to be confirmed with long-term follow-up studies. Therefore, the effects of SGLT2i on prognosis should be evaluated in a prospective randomized controlled trial using imaging modalities and a long-term follow-up period

The use of SGLT2 inhibitors not only improves fatty liver and fibrosis, but also inhibits various cancers, including liver cancer. In addition, since SGLT2 inhibitors are existing drugs, they are efficient in terms of healthcare economics, and their use is expected to expand in the future.

### 4.2. Inhibition of Liver Fibrosis Using Cell Transplantation

We have conducted basic research as well as clinical studies on the inhibition of liver fibrosis and hepatocarcinogenesis through cell transplantation therapy for liver cirrhosis, which is the most important risk factor in the development of HCC. The treatment of liver cirrhosis aims to improve liver functional reserves and inhibit HCC. In 1997, Asahara et al. reported the presence of bone marrow-derived endothelial progenitor cells (EPC) within a subset of human peripheral blood mononuclear cells and further demonstrated that these cells are enriched in the CD34-positive cell fraction [134]. The most important feature of these cells is that they migrate selectively to injured areas due to inflammation or ischemia and contribute to angiogenesis in adults, as well as in fetal embryonic stages [135].

We have investigated the therapeutic effects of human peripheral blood CD34-positive cells on a rat cirrhosis model and found these cells inhibited liver fibrosis, promoted liver regeneration, improved portal hypertension, and improved prognosis by promoting intrahepatic angiogenesis and supplying various growth factors produced by EPCs to the surrounding tissues [136,137]. Furthermore, we performed a multicenter, randomized, open-label, comparative study to investigate the safety and efficacy of transhepatic arterial administration of granulocyte-colony stimulating factor (G-CSF) mobilized autologous peripheral blood CD34-positive cells compared with standard therapy in patients with HCV-related decompensated liver cirrhosis [138]. In terms of efficacy, at week 24 post-enrolment, the improvement rate in Child–Pugh score, which evaluates liver functional reserve using a scoring system based on serum albumin, total bilirubin levels and prothrombin time–international normalized ratio (PT-INR), of one or more points was superior in the CD34-positive cell transplant group (50% vs. 25% in the standard treatment group), and 40% of patients in the CD34-positive cell transplant group exhibited an improvement from decompensated to compensated cirrhosis, whereas all patients in the standard treatment group remained in decompensated cirrhosis. Regarding safety, there were no serious adverse events, no deaths or all-cause mortality due to liver cirrhosis, and no development of HCC. Recently, given the increasing number of patients with MASH-derived cirrhosis, we evaluated the efficacy of CD34-positive cell transplantation in a murine MASH model. The results show that CD34-positive cell transplantation was effective in MASH cirrhotic mice [139]. Clinical trials targeting human subjects suffering from MASH-related cirrhosis are scheduled to begin in the near future.

### 4.3. Exercise and Myokines

Sarcopenia is an independent risk factor for MASLD and HCC [140,141,142]. Skeletal muscles are not only responsible for physical activity, but are also deeply involved in the regulation of various metabolisms, including the glucose and lipid metabolisms. We have performed a systematic review and found that both aerobic and resistance exercises improve MASLD [143]. However, these two forms of exercise have distinct characteristics. The intensity and energy expenditure are significantly lower for resistance exercise compared to aerobic exercise. These results suggest that resistance exercise improves MASLD by mechanisms other than exercise-induced energy expenditure.

Skeletal muscle is known as an endocrine organ. Through muscle contraction, myocytes release small peptides and cytokines called myokines [144]. IL-6 is a first-identified myokine by Pedersen et al. [145], and resistance exercise combined with aerobic exercise has been reported to increase serum IL-6 levels, leading to an improvement in insulin resistance as well as MASLD [144,146]. Tsutsui et al. recently reported that exercise promoted the secretion of IL-15 from skeletal muscle and IL-15 improved MASLD through the suppression of the accumulation of liver bone-marrow-derived macrophages and programmed death receptor-1+ CD8+ T cells [147]. Thus, resistance exercise improves MASLD partly by an induction of myokines including IL-6 and IL-15.

We have developed a low-intensity 10 min resistance exercise program for the prevention and treatment of MASLD based on the results of a systematic review of the effects of exercise on MASLD [148]. The resistance exercise program has been reported to increase the serum level of granulocyte-colony stimulating factor (G-CSF) and to decrease the serum level of interferon-gamma-induced protein-10 (IP-10) and platelet-derived growth factor-BB (PDGF-BB), and to subsequently improve hepatic fibrosis in patients with MASLD [149]. These findings suggest that resistance exercise may suppress hepatic fibrosis and its complications by regulation of chemokines/cytokines. In fact, a meta-analysis reported that resistance exercise reduces the incidence of serious events in patients with liver cirrhosis [150].

The serum levels of myokines are associated with the survival of patients with HCC. Furthermore, several studies have demonstrated that exercise suppresses the onset of HCC and improves the prognosis of patients with HCC [151,152,153]. Myostatin and decorin are myokines that exert antiproliferative effects against cancers and antiangiogenic properties in vitro. We have previously reported that myostatin and decorin serum levels are associated with overall survival in patients with HCC [154,155]. Fractalkine/CX3CL1 is also a myokine which regulates tumor microenvironments by inducing anti-tumor immunity in a mouse model of HCC [156]. A high expression of Fractalkine/CX3CL1 has been reported to have significantly fewer intra- and extrahepatic recurrences of HCC and a significantly better prognosis in patients with HCC [157]. The low-intensity 10 min resistance exercise program we developed is associated with the upregulation of serum fractalkine/CX3CL1 levels [148]. Thus, these findings suggest that even low-intensity exercise may alter myokine expressions in humans, suppress HCC, and improve the prognosis of patients with HCC.

Future research should encompass both basic and clinical aspects. Basic research should investigate metabolic, inflammatory, and immune pathways to elucidate the molecular mechanisms linking MASLD to hepatocarcinogenesis. Clinically, efforts should focus on identifying reliable biomarkers for early HCC detection and risk stratification, evaluating the long-term effects of anti-fibrotic and metabolic therapies, and integrating multi-omics approaches to refine HCC prevention strategies.

## 5. Conclusions

In conclusion, in this review, we summarize the recent progress in the pathogenic mechanisms of MASLD-associated HCC. In addition, we introduce new therapeutic strategies for preventing the development of HCC based on the pathogenesis of the disease. Understanding the latest pathogenesis of HCC is crucial in developing new therapeutic strategies for HCC. It is important to focus on novel insights and perspectives that will advance this field in both basic and clinical aspects.

## Figures and Tables

**Figure 1 cells-14-00428-f001:**
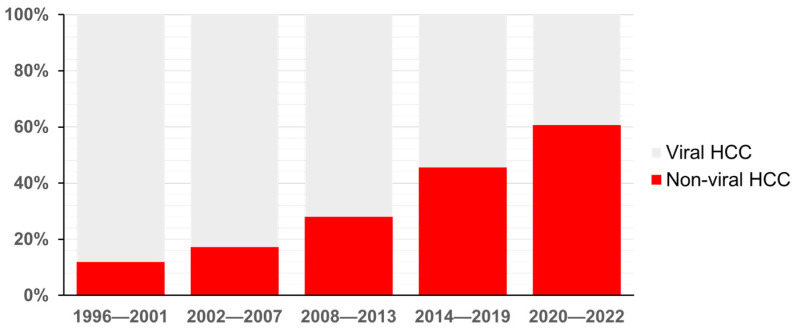
Changes in the percentage of non-viral HCC in Japan. The percentage of non-viral HCC has dramatically increased. These data from 1996 to 2019 are adopted from [4].

**Figure 2 cells-14-00428-f002:**
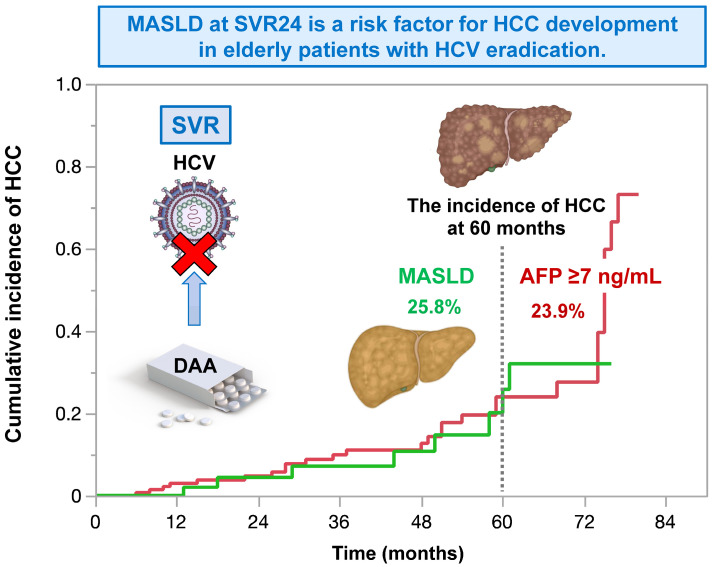
In elderly patients with HCV eradication, MASLD at 24 weeks of sustained virological response increases the risk for HCC. The impact of MASLD on the risk for HCC is comparable to that of alpha-fetoprotein ≥ 7 ng/mL. This figure is adopted from [76].

**Figure 3 cells-14-00428-f003:**
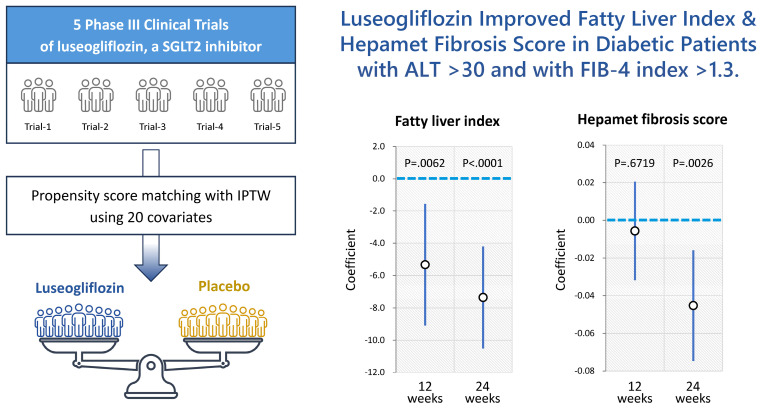
A pooled meta-analysis revealed that 24-week treatment with luseogliflozin ameliorates hepatic steatosis and fibrosis indexes in patients with type 2 diabetes melllitus, especially those with liver injury. This figure is adopted from [116].

**Figure 4 cells-14-00428-f004:**
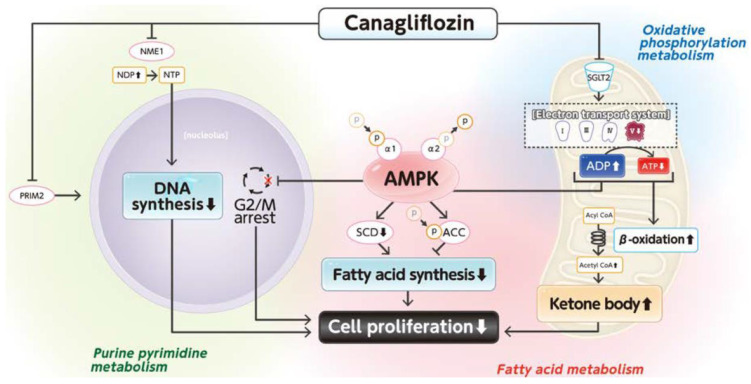
Mechanisms for the canagliflozin-induced suppression of cell proliferation in a human hepatoma cell line. This figure is adopted from [131].

**Figure 5 cells-14-00428-f005:**
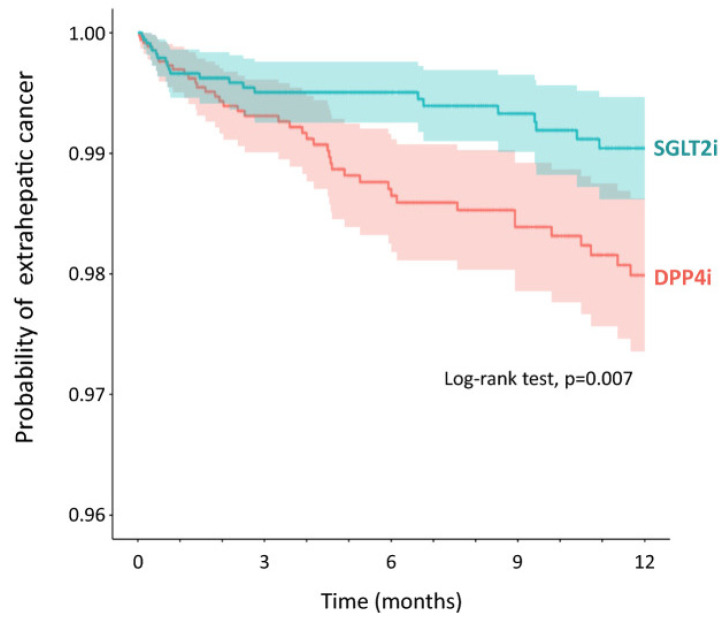
The incidence of extrahepatic cancer in the SGLT2i group compared to the DPP4i group. The incidence of extrahepatic cancer is significantly lower in the SGLT2i group compared to that of the DPP4i group. This figure is adopted from [102].

**Table 1 cells-14-00428-t001:** Mechanisms of MASLD-related HCC.

Mechanisms of MASLD-Related HCC
Lipid accumulation-induced chronic inflammation
Oxidative stress and DNA damage
Activation of immune cells including macrophages and lymphocytes
Hepatic fibrosis
Metabolic dysfunction-related inflammation and carcinogenesis
Liver microenvironment and pro-carcinogenic signaling pathways

## Data Availability

No new data was created or analyzed in this study.

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
