# Peer review of "Pathogenic Mechanisms of Metabolic Dysfunction-Associated Steatotic Liver Disease (MASLD)-Associated Hepatocellular Carcinoma"

_cells, 2025, doi:10.3390/cells14060428_

Round 1

Reviewer 1 Report

Comments and Suggestions for Authors

·         This study requires restructuring because of redundant sentences that require consolidation. Additionally, findings from similar investigations should be grouped together for better coherence and analysis.

·         In line 56 what did you mean by the number 2 in this sentence? (In addition, steatotic liver disease (SLD) is the most common cause of chronic liver disease, with an estimated worldwide prevalence of 2)

·         Many abbreviations have not been explained or their meanings mentioned, such as MFLAD , NASH , SGLT2i and DPP4i

·         This sentence needs to be paraphrased (several large randomized clinical trials even in patients with no type 2 diabetes mellitus).

·         You did not mention why luseogliflozin increased serum alanine aminotransferase compared to those in the control group.

·         In line 368, you wrote fetal. I have suggested that you refer to the fetuses.

·         This manuscript requires further English editing.

Comments on the Quality of English Language

 This manuscript requires further English editing.

Author Response

Thank you very much for your letter regarding our manuscript (cells-3409054). We sincerely thank you for taking your valuable time to review this paper. We also appreciate your comment, which has helped us to improve our manuscript. In line with your comment, please find below our point-by-point response.

Comments and Suggestions for Authors

Comments 1: This study requires restructuring because of redundant sentences that require consolidation. Additionally, findings from similar investigations should be grouped together for better coherence and analysis.

Response 1: Thank you for your comments. We have deleted redundant sentences according to your suggestions. We also have grouped findings from similar investigations for better coherence and analysis. Again, we appreciate your comment, which has helped us to improve our manuscript.

Comments 2: In line 56 what did you mean by the number 2 in this sentence? (In addition, steatotic liver disease (SLD) is the most common cause of chronic liver disease, with an estimated worldwide prevalence of 2).

Response 2: We appreciate your careful proofreading and apologize for our typo. The last part of the sentence was deleted. We have revised the sentence as follows: Furthermore, steatotic liver disease (SLD) is the leading cause of chronic liver disease, affecting approximately 25% of the global adult population (line 61).

Comments 3: Many abbreviations have not been explained or their meanings mentioned, such as MFLAD, NASH, SGLT2i and DPP4i.

Response 3 : Thank you for your comments. We apologize that we did not spell out many abbreviations. We have explained the abbreviations and revised the manuscript according to your suggestion.

Comments 4: This sentence needs to be paraphrased (several large randomized clinical trials even in patients with no type 2 diabetes mellitus).

Response 4: We appreciate your comment. The sentence has been paraphrased as follows: In addition to its glucose-lowering properties, SGLT2 inhibitors have shown cardiovascular and renal benefits in numerous large-scale randomized clinical trials, even among patients without type 2 diabetes mellitus (lines 346-349).

Comments 5: You did not mention why luseogliflozin increased serum alanine aminotransferase compared to those in the control group.

Response 5: We appreciate your careful proofreading. The “increased” is a typo and has been corrected by “decreased” as follows: They found that luseogliflozin suppressed hepatic lipid accumulation and decreased serum alanine aminotransferase compared to those in the control group (line 356).

Comments 6: In line 368, you wrote fetal. I have suggested that you refer to the fetuses.

Response 6: We deeply appreciate your careful proofreading. We have revised “fetal” to “fetal embryonic stages” in the revised manuscript (line 471).

Comments 7: This manuscript requires further English editing.

Response 7: We appreciate your careful proofreading. We have performed English editing throughout the manuscript.

Reviewer 2 Report

Comments and Suggestions for Authors

This is a timely review on an important topic. A few suggestions to the authors are found below.

Major:

-Based on the authors’ analysis of current evidence, future research directions, both basic and clinical, must be highlighted for each section and explicitly stated in the Conclusion. Focus on novel insights and perspectives that pushes the field forward.

-Therapeutic development starting with preclinical mechanistic in vitro and in vivo studies of candidate compounds is clearly needed since only one drug is approved so far. Mention these candidate compounds, approaches to identify them, and/or potential molecular targets to exploit.

-A critical analysis of previous work is lacking. In many instances, the authors report the work of others without highlighting weaknesses or limitations. This must be revised to better contextualize the cited references.

Minor:

-Abstract: “Sodium-glucose cotransporter-2 inhibitor (SGLT2 inhibitor), an anti-diabetic medication”. SGLT inhibitors is a class of medications; it is not a medication (drug). Please correct.

-“This data from 1996 to 1019” – 2019?

-Please make sure permissions to use the figures have been obtained from all copyright holders.

-“Lipid accumulation-caused” – lipid accumulation-induced.

-“Immune-related cells” – immune cells.

-Please refine the style and writing of the manuscript since some sentences seem off.

Author Response

Thank you very much for your letter regarding our manuscript (cells-3409054). We sincerely thank you for taking your valuable time to review this paper. We also appreciate your comment, which has helped us to improve our manuscript. In line with your comment, please find below our point-by-point response.

Major:

Comments 1: Based on the authors’ analysis of current evidence, future research directions, both basic and clinical, must be highlighted for each section and explicitly stated in the Conclusion. Focus on novel insights and perspectives that pushes the field forward.

Response 1: We deeply appreciate your understanding of our manuscript. We have added descriptions for future directions at the end of each section. We also added the description in the Conclusion. Thank you for taking your valuable time to review our manuscript.

Comments 2: Therapeutic development starting with preclinical mechanistic in vitro and in vivo studies of candidate compounds is clearly needed since only one drug is approved so far. Mention these candidate compounds, approaches to identify them, and/or potential molecular targets to exploit.

Response 2: We understand the importance of mentioning candidate compounds. Since this is an invited review article based on our previous reports, we briefly introduce the candidate compounds for MASLD as follows: Recently, resmetirom, a thyroid hormone receptor-β agonist, was approved in the USA [1]. In addition, the following compounds are under clinical trials and are good candidates for MASLD: GLP-1 receptor agonist, a dual agonist of glucagon receptor and GLP-1 receptor, pan-PPAR agonist, and FGF21 analogue [2-11]. However, resmetirom are not available elsewhere and these new candidates have not been approved yet. Therefore, it is necessary to take measures to prevent the onset and progression of HCC.

Comments 3: A critical analysis of previous work is lacking. In many instances, the authors report the work of others without highlighting weaknesses or limitations. This must be revised to better contextualize the cited references.

Response 3: We appreciate your valuable comment. We highlighted the weaknesses or limitations as follows: This study has several limitations. Firstly, all phase III clinical trials were conducted in Japan. Secondly, the evaluation of hepatic steatosis and fibrosis relied on non-invasive biological tests. Thirdly, the treatment duration was 24 weeks. Therefore, future research should be structured as a meta-analysis of clinical trials conducted in various regions, incorporating data from imaging modalities or liver biopsies, and extending the treatment periods (lines 372-377).

Again, we appreciate your comment, which has helped us to improve our manuscript. In line with your comment, please find below our point-by-point response.

Minor:

Comments 1: Abstract: “Sodium-glucose cotransporter-2 inhibitor (SGLT2 inhibitor), an anti-diabetic medication”. SGLT inhibitors is a class of medications; it is not a medication (drug). Please correct.

Response 1: Thank you for your careful proofreading. As you pointed out, SGLT inhibitors is a class of medications; it is not a medication. We have corrected the sentence as follows: Sodium-glucose cotransporter-2 (SGLT2) inhibitor, a class of medications, has been reported to exert anti-tumor effects on HCC by regulating metabolic reprogramming (line 28).

Comments 2: “This data from 1996 to 1019” – 2019?

Response 2: Thank you for pointing out. We have revised “1019” to “2019” in the revised manuscript according to your suggestions (line 72).

Comments 3: Please make sure permissions to use the figures have been obtained from all copyright holders.

Response 3: Thank you for your kindness. We have obtained the permission to use the figures from all copyright holders.

Comments 4: “Lipid accumulation-caused” – lipid accumulation-induced.

Response 4: Thank you for your comments. We have corrected “lipid accumulation-caused” to “lipid accumulation-induced” in the revised manuscript according to your suggestions (Table 1 and line 167).

Comments 5: “Immune-related cells” – immune cells.

Response 5: Thank you for your comments. We have corrected “immune-related cells” to “immune cells” in the revised manuscript according to your suggestions (Table 1).

Comments 6: Please refine the style and writing of the manuscript since some sentences seem off.

Response 6: We appreciate your careful proofreading. We have revised inappropriate sentences in the revised manuscript.

Reviewer 3 Report

Comments and Suggestions for Authors

In their manuscript, authors described the connection and progression between metabolic dysfunction-associated steatotic liver disease (MASLD) and hepatocellular carcinoma (HCC).

The manuscript is well written and I have not revealed necessity of linguistic corrections.

However, I have some suggestions and questions for the authors.

First, HCC is classified in several ways based on structure, mutation, metabolism, immuno-cell population and other conditions (some examples are: 10.1016/bs.acr.2020.10.001; 10.1053/j.gastro.2017.06.007; 10.3389/fimmu.2022.1010554). In my opinion, authors should describe the progression focusing on HCC subclass or they should indicate what kind of HCC subgroup were taken in consideration for their review. Furthermore, and this could be considered as the second point, authors should indicate the inclusion/exclusion criteria for patients used to describe the novelty of their manuscript.

Third, authors used only alpha fetoprotein (AFP) as cutoff for HCC risk factor. However, it was well described that only AFP is not a good marker for HCC, in fact, doctors used images, biomarkers and other techniques to improve the AFP biomarker for HCC diagnosis. Have authors considered other data to describe the HCC risk?

Finally, authors described reactive oxygen species (ROS) and reactive nitrogen species (RNS). They are produced from different immuno-cell populations, in particular RNS are principally produced by activated neutrophils, and they have a crucial role in the MASLD arisen and progression (10.1186/s12876-024-03394-6). I think that this part should be described more in detail due to its role in the inflammatory liver.

Author Response

To Reviewer 3

Thank you very much for your letter regarding our manuscript (cells-3409054). We sincerely thank you for taking your valuable time to review this paper. We also appreciate your comment, which has helped us to improve our manuscript. In line with your comment, please find below our point-by-point response.

Comments 1: First, HCC is classified in several ways based on structure, mutation, metabolism, immuno-cell population, and other conditions (some examples are: 10.1016/bs.acr.2020.10.001; 10.1053/j.gastro.2017.06.007; 10.3389/fimmu.2022.1010554). In my opinion, authors should describe the progression focusing on HCC subclass or they should indicate what kind of HCC subgroup was taken in consideration for their review.

Response 1: We appreciate your valuable comment. As you pointed out, HCC is classified in several ways based on structure, mutation, metabolism, immuno-cell population, and other conditions [1-3]. Following your suggestion, we have added a paragraph for the HCC subclass that highlights the complex interactions between chronic metabolic inflammation, immune suppression, and tumor progression in MASLD-derived HCC in the revised manuscript (lines 122-159). Again, we appreciate your comment, which has helped us to improve our manuscript. In line with your comment, please find below our point-by-point response.

Comments 2: Furthermore, and this could be considered as the second point, authors should indicate the inclusion/exclusion criteria for patients used to describe the novelty of their manuscript.

Response 2: Thank you for your comment. In the revised manuscript, we have added the description of inclusion and exclusion criteria for patients.

Comments 3: Third, the authors used only alpha-fetoprotein (AFP) as the cutoff for HCC risk factor. However, it was well described that only AFP is not a good marker for HCC, in fact, doctors used images, biomarkers, and other techniques to improve the AFP biomarker for HCC diagnosis. Have authors considered other data to describe the HCC risk?

Response 3: Thank you for your comments. We totally agree with your comment. Images and biomarkers are useful in improving the diagnostic ability of AFP. We have added the following sentences at the end of the paragraph: In the future, even when viral hepatitis is already under control, it is important to be aware of the complications of MASLD and to continue to manage it with images and biomarkers (lines 315-317).

Comments 4: Finally, the authors described reactive oxygen species (ROS) and reactive nitrogen species (RNS). They are produced from different immuno-cell populations, in particular, RNS are principally produced by activated neutrophils, and they have a crucial role in the MASLD arisen and progression (10.1186/s12876-024-03394-6). I think that this part should be described more in detail due to its role in the inflammatory liver.

Response 4: Thank you for your valuable comments. We have revised our manuscript to address your suggestion, explaining ROS and RNS in more detail by citing the indicated article [4] as described below (lines 130-138): ROS and RNS are produced by different immune cell populations in response to liver inflammation [5]. Notably, RNS are predominantly generated by activated neutrophils, which play a crucial role in the onset and progression of MASLD [4]. The excessive production of RNS amplifies oxidative stress, further exacerbating cellular damage and perpetuating the cycle of inflammation and injury. This process not only accelerates the progression of MASLD but also creates a microenvironment favorable for hepatocarcinogenesis. By inducing oxidative modifications in DNA, proteins, and lipids, RNS contributes to the chronic liver damage that underpins the transition from MASLD to HCC [6, 7].

References

  1. Sia, D.; Jiao, Y.; Martinez-Quetglas, I.; Kuchuk, O.; Villacorta-Martin, C.; Castro de Moura, M.; Putra, J.; Camprecios, G.; Bassaganyas, L.; Akers, N.; et al. Identification of an Immune-specific Class of Hepatocellular Carcinoma, Based on Molecular Features. Gastroenterology. 2017, 153, 812-26. doi: 10.1053/j.gastro.2017.06.007
  2. Chidambaranathan-Reghupaty, S.; Fisher, P.B.; Sarkar, D. Hepatocellular carcinoma (HCC): Epidemiology, etiology and molecular classification. Adv Cancer Res. 2021, 149, 1-61. doi: 10.1016/bs.acr.2020.10.001
  3. Xu, W.; Nie, C.; Lv, H.; Chen, B.; Wang, J.; Wang, S.; Zhao, J.; He, Y.; Chen, X. Molecular subtypes based on Wnt-signaling gene expression predict prognosis and tumor microenvironment in hepatocellular carcinoma. Front Immunol. 2022, 13, 1010554. doi: 10.3389/fimmu.2022.1010554
  4. Lu, Y.; Xu, X.; Wu, J.; Ji, L.; Huang, H.; Chen, M. Association between neutrophil-to-high-density lipoprotein cholesterol ratio and metabolic dysfunction-associated steatotic liver disease and liver fibrosis in the US population: a nationally representative cross-sectional study using NHANES data from 2017 to 2020. BMC Gastroenterol. 2024, 24, 300. doi: 10.1186/s12876-024-03394-6
  5. Allameh, A.; Niayesh-Mehr, R.; Aliarab, A.; Sebastiani, G.; Pantopoulos, K. Oxidative Stress in Liver Pathophysiology and Disease. Antioxidants (Basel). 2023, 12. doi: 10.3390/antiox12091653
  6. Martinez, M.C.; Andriantsitohaina, R. Reactive nitrogen species: molecular mechanisms and potential significance in health and disease. Antioxid Redox Signal. 2009, 11, 669-702. doi: 10.1089/ars.2007.1993
  7. Li, S.; Tan, H.Y.; Wang, N.; Zhang, Z.J.; Lao, L.; Wong, C.W.; Feng, Y. The Role of Oxidative Stress and Antioxidants in Liver Diseases. Int J Mol Sci. 2015, 16, 26087-124. doi: 10.3390/ijms161125942

Round 2

Reviewer 2 Report

Comments and Suggestions for Authors

The authors have adequately addressed my comments.

Reviewer 3 Report

Comments and Suggestions for Authors

Authors have done a very meticulous and commendable work with their revision. They have fixed all the issues raised in the review, and the manuscript is now more detailed and robust. Based on my opinion, I would like to recommend accepting the manuscript in its present form.